# Psychological Responses According to Gender during the Early Stage of COVID-19 in Spain

**DOI:** 10.3390/ijerph18073731

**Published:** 2021-04-02

**Authors:** Lucía del Río-Casanova, Milagrosa Sánchez-Martín, Ana García-Dantas, Anabel González-Vázquez, Ania Justo

**Affiliations:** 1Department of Psychiatry, University of Santiago de Compostela, Santiago de Compostela, 15705 Santiago de Compostela, Spain; ludelriocasanova@gmail.com; 2Department of Psychology, School of Social Sciences and Humanities, Universidad Loyola Andalucía, Dos Hermanas Campus, 41014 Sevilla, Spain; agdantas@uloyola.es or; 3Dantas Psicología Clínica (Private Therapy Clinic), 41011 Sevilla, Spain; 4A Coruña University Hospital, University of Santiago de Compostela, Santiago de Compostela, 15003 A Coruña, Spain; anabel.gonzalez.vazquez@gmail.com; 5Imaya Medical Institute (Private Therapy Clinic), 36204 Vigo, Spain; aniajusto@gmail.com

**Keywords:** COVID-19 confinement, gender, mental health, psychological impact, quarantine consequences, post-traumatic stress

## Abstract

Background: Current research has pointed out an increased risk of mental health problems during the COVID-19 pandemic in women compared to men, however the reason for this difference remains unclear. The aim of this research is to study early psychological responses to the pandemic in the Spanish general population, focusing on gender differences. Methods: Nine to 14 days after the declaration of a state of emergency an online survey was conducted assessing sociodemographic, health, behavioral and COVID-19-related variables. Mental health status was evaluated by the Depression, Anxiety and Stress Scale (DASS-21), the Impact of Event Scale-Revised (IES-R), and the Self-Care Scale (SCS). Results: The study included 3520 respondents: 2611 women and 909 men. Women scored significantly higher in DASS-21 and IES-R (*p* < 0.05) and were more likely to somatize, suffer from hypochondriasis, sleeping disturbances and claustrophobia (*p* < 0.05). Being a woman can be considered a risk factor for intrusive thoughts, avoidance mechanisms, stress and anxiety (*Odd Ratio* = 2.7/2.3/2.3/1.6). The risk of presenting posttraumatic symptoms and emotional distress was greater in women (*Odd Ratio* = 6.77/4.59). General linear models to predict IES-R and DASS-21 scores clarified which variables were gender specific, such as main concerns. Conclusions: This study provides evidence that at early stages of the pandemic, women mental health was more impacted and that both genders show different concerns. Gender perspective in secondary and tertiary prevention strategies must be taken into account when facing the distress associated with the pandemic.

## 1. Introduction

The World Health Organization (WHO) declared a global pandemic on 11 March 2020. The magnitude of the situation and the quarantine measures in many countries have no precedents, and understanding how these circumstances influence mental health is of special interest.

The pandemic has particularly hit the mental health of health care workers [1], COVID-19 survivors [2,3] and their families, as well as people with mental disorders before the pandemic [4,5]. When focusing on general population, most studies found moderate to severe depressive, stress and/or anxiety symptoms in a large percentage (16–64%) of the population studied [6,7,8,9,10] while post-traumatic symptoms have been reported in 7–53% [6,11].

Diverse individual and sociodemographic factors have been linked to worse mental health during the COVID-19 pandemic [12]. The perception of uncertainty [13] and the levels of isolation [14] are linked to higher risk of negative psychological responses. Women, young, and uneducated people have presented worse psychological response in different studies [6,11,15], as they are also subgroups with poorer neighborhood relationships, poorer self-perceived health, and higher economic impact [15]. Concerning the age factor, results are diverse: worse psychological responses in elderly people [16], worse responses on the two extremes (younger and older participants) [17], no age differences [6,18], or worse responses for young people [8].

Concerning gender, although several studies have reported that COVID-19 is deadlier for infected men than women with a 2.8% fatality rate in men versus 1.7% in women [19], most literature points at a vulnerability factor linked to female gender concerning mental health during the pandemic [13]. Women have been identified as a risk group for suffering a stronger deterioration in their mental health during the COVID-19 pandemic, with significant increased stress, anxiety, and depression rates when compared to men [6,7,8,17]. In the systematic review conducted by Vindegaard, a gender mental health vulnerability in women was found in most of the 43 studies revised [12], including general distress measures, anxiety, depression, insomnia, and posttraumatic symptoms. However, there is a lack of research focused specifically in gender. Liu et al. is the only study conducted with this aim [11]. Liu found that the prevalence of posttraumatic stress symptoms (PTSS) in the worst-affected areas of China one month after the COVID-19 outbreak was 7%, with women reporting significantly higher PTSS in the domains of re-experimentation, negative disturbances in cognition or mood, and hyperarousal [11]. In Spain, González-Sanguino et al. found higher rates of all the mental health symptoms evaluated in women when compared to men, and the regression model conducted considered gender as a predictive variable for anxiety and post-traumatic symptoms but not for other psychological symptoms [11]. 

Generally, the gender issue has been marginally addressed and has not been considered the main objective of the studies, and because of that, we do not have any explanatory models in this regard and there is also a lack of guidelines for gender adjusted psychological interventions. Current literature analyzes gender as one more variable linked to a higher percentage of mental health issues in women when compared to men. However, this is not only a question of quantity but also of quality. Are both genders concerned about the same issues? Which variables can predict those gender differences in mental health? How can psychological interventions adjust to an evidence-based gender analysis? These issues remain unclear. In this sense, the aim of this research is to study the role of gender on mental health responses during the early days of the COVID-19 pandemic in the Spanish general population. Based on the results, specific recommendations are postulated.

Finally, the evolution in the population mental health along the pandemic is also an important issue. Some longitudinal studies have been recently published and confirm the deterioration in the population mental health comparing pre-pandemic to pandemic data [20,21,22]. Other two longitudinal studies have shown that this deterioration persisted in several measures in the following months and that gender differences remained [23,24]. There is a lack of longitudinal research focused on gender along the pandemic. Taking this into account, the analysis of men and women mental health status at the early stages of the pandemic, could be useful in order to propose gender adjusted interventions that may decrease the impact of the pandemic and have positive effects along time. 

## 2. Materials and Methods

### 2.1. Procedures and Instruments

The inclusion criteria were determined as follows: (a) being 18 years of age and older and (b) living in Spain. The general population was approached by social media and encouraged to participate with no economic reward for participation. A snowball sampling strategy was used through an online survey. Data were recruited from 23 March to 28 March (2020), 9 to 14 days after the declaration of the state of emergency above a general quarantine. A large increase in infected and death rates took place during that time (from 2182 deaths to 5690 deaths) [25].

The following data were collected: (1) sociodemographic data: gender, age, employment, and education; (2) health variables: perceptions of physical and mental state, antecedents of psychiatric illness, and some symptoms (such as somatization, agoraphobia, sleep patterns, and hypochondriac concerns); (3) behavioral variables: level of isolation, general routines, and toxic habits during the quarantine; (4) variables linked to the COVID-19 pandemic: housing and household characteristics during the pandemic, main concerns regarding COVID-19, different measures of infection and contact and perceived risk of COVID-19.

Mental health status was assessed by the Depression, Anxiety, and Stress Scale (DASS-21) [26,27], with 21 items divided into three subscales: depression, anxiety, and stress. The depression subscale scores were categorized following previous research [6] into normal (0–9), mild (10–12), moderate (13–20), severe (21–27), and extremely severe (28–42). The anxiety subscale scores were categorized into normal (0–6), mild (7–9), moderate (10–14), severe (15–19), and extremely severe (20–42). The stress subscale scores were categorized into normal (0–10), mild (11–18), moderate (19–26), severe (27–34), and extremely severe (35–42).

The Impact of Event Scale-Revised (IES-R) [28,29] provides an indirect measure of symptoms related to posttraumatic disorder, composed of 22 items with three subscales: avoidance, intrusion, and hyperarousal. To categorize this variable, cut points for the IES-R were selected following previous research [30]: 0–23 (normal), 24–32 (mild), 33–36 (moderate), and >37 (severe).

For our study, the Cronbach’s alpha index was 0.94 for the SCS, 0.91 for the IES-R and 0.86, 0.85 and 0.90 for the depression, anxiety and stress subscales, respectively, of the DASS-21.

Self-care patterns of the participants were assessed by the Self-Care Scale (SCS) [31]. This scale is composed of 31 items and divided into 6 subscales: self-destructive behavior, taking into account needs of oneself, resentment over not receiving reciprocity, difficulty in receiving and accepting help, lack of tolerance of shared positive affect and absence of positive activities. 

### 2.2. Statistical Analysis

First, descriptive statistics were calculated (percentages, mean and standard deviation), depending on the variable nature, qualitative or quantitative. Later, we compared men and women respect to different variables. Concretely, to analyze differences between groups respect to sociodemographic variables (measured as categorical) we used Chi-square test with the McNemar test correction for 2 × 2 tables and the Bonferroni correction for multiple testing. Later, we used Student’s t-tests for analyzing if measures of mental health (measured as quantitative) were significantly different in men and women. We adjusted for multiple comparison applying the Bonferroni correction and we used the robust Welch’s Test whether the assumption of homogeneity of variances (Levene’s Test) was violated. All these analyses were complemented with the corresponding effect size statistic, either directly obtained from the software or calculated by an online calculator [32]. Cramer’s V was interpreted as 0.10, 0.30, and 0.50 for small, medium, and large effects, respectively; and Cohen’s d considering values of 0.2, 0.5, and 0.8 as small, medium, and large sizes, respectively [33].

With the objective of obtaining the risk associated with being a man or woman (odds ratio) in predicting individual psychological variables, we executed simple linear regressions, one by each dependent variable and with gender as the unique independent variable.

Finally, four univariate general linear models were calculated for men and women separately to determine which variables predicted IES-R and DASS-21 scores in each group. We analyzed the homogeneity variance assumption, and we applied the correction of Bonferroni for multiple testing when the assumption of homogeneity variance was accepted or the correction of Games–Howell when the assumption was violated. We used the partial eta-square (*ŋ^2^_p_*) as the effect size statistic [33]; the reference values were 0.01, 0.06 and >0.14 for small, medium, and large sizes, respectively.

In general, results are shown with a significance level of *p* < 0.05, and all tests were two-tailed. For multiple T-test we applied the Bonferroni correction, considering a level of significance of 0.00416 for rejecting the null hypothesis. For Chi-square test the Bonferroni correction is directly applied over the significance value that the software report.

Statistical analysis was performed using SPSS Statistics 26.0 (IBM SPSS Statistics, New York, NY, USA).

## 3. Results

### 3.1. Survey Respondents

The survey was linked to an email address, so participants could not access the platform twice. In order to avoid random answering patterns, we checked carefully the data base, analyzing patterns of responses, possible missing data, and outliers.

During the recruitment period, 4139 people completed the survey. We removed 120 subjects because they did not provide informed consent, 21 people were under 18 years old, 430 were respondents from other countries, 4 participants perceived themselves to be included in other gender categories (the small number of participants impeded our ability to analyze this specific subgroup), and 44 abnormal responses (the more common was to provide the same response, for example 3, in all questions). A final sample of 3520 was included. 

Most participants were women (74.2%), with a mean age of 39.24 years (*SD* = 12). Concerning housing, only 18.2% lived in a house with outdoor areas, 26.2% in a flat without any outdoor areas, 43.9% in a flat with either a terrace or balcony, 9.9% in a cottage or a detached house with garden, and 1.8% in other type of residence. Respect to whether participants changed their residence in the last days, 85.1% always have lived at the same house and only 14.9% moved to another house, concretely, 6.9% changed their residence because the Coronavirus, 4.7% moved to other residence because other reason and 3.3% did not provide the information. Regarding household characteristics, 31.6% of the participants were living with their partners and children, 24.1% lived as a couple, 15.3% lived with their parents, 10.4% lived alone, 4.1% lived with their partners, children, and other family member, 0.1% lived with their parents-in-laws, and 14.4% lived with other combination of people or animals. 

#### 3.1.1. Sociodemographic Characteristics

The most relevant sociodemographic characteristics analyzed with respect to gender can be seen in Table 1. Those characteristics that reached statistically significance between men and women (corrected standardized residual > ±1.96) were highlighted. 

#### 3.1.2. Mental Health and Psychological Impact

As we can see in Table 2, women showed significantly poorer mental health status, with more anxiety, depression, and stress scores than men. For more visual information, the categorized scores of the DASS-21 on the depression and anxiety subscales are shown in Figure 1.

Women reported higher scores than men, but non-significantly difference, regarding self-care patterns (*t*(1) = −2.14, *p* > 0.004, *d* = 0.08) and no positive activities subscale (*t*(1) = −2.19, *p* > 0.004, *d* = 0.08). On the other hand, significantly differences between groups were found on self-destructive behaviors subscale (*t*(1) = −4.96, *p* < 0.001, *d* = 0.19).

A total of 21% of the men vs. 36% of women had been previously diagnosed with a mental disorder (*X^2^*(2) = 73.01, *p* < 0.001, *V* = 0.14). Amongst those with a diagnosis, 11.5% of men and 21.7% of women reported a worsening in their symptoms during the state of alarm, reaching statistical significance (*X^2^*(2) = 32.15, *p* < 0.01, *V* = 0.15).

When asked for their subjective perception of their general mental state, women considered themselves less healthy than men did (*t*(1) = 8.55, *p* < 0.001, *d* = 0.19).

Other psychiatric symptoms, such as somatization, agoraphobia, hypochondriasis and sleep disturbances, were assessed for both men and women, with women being the most symptomatic group for the 4 variables (see Table 3).

#### 3.1.3. Behavioral Variables

More women than men took medication in the same way or more than usual, while more men than women did not take medication (*X^2^*(3) = 25.05, *p* < 0.01, *V* = 0.08). In relation to alcohol, more women than men stated that they do not usually drink alcohol (*X^2^*(2) = 110.68, *p* < 0.001, *V* = 0.18), and the same occurred with drugs. However, at the same time, more men than women said that they had decreased their drug use during the quarantine (*X^2^*(2) = 31.41, *p* < 0.001, *V* = 0.09).

#### 3.1.4. Variables Linked to the COVID-19 Pandemic: Main Concerns

More men than women thought that they were not infected (73.1% versus 66.1%; *X^2^*(2) = 15.26, *p* < 0.001, *V* = 0.06). Consistently, 97.7% of men versus 95.6% of women had not been quarantined for suffering from symptoms compatible with COVID-19 infection (*X^2^*(1) = 7.72, *p* < 0.001, *V* = 0.04).

Concerning the level of isolation during the quarantine period, more women than men stated that they stayed at home the whole day, while more men than women went out only for essential reasons (work, shopping, walking their dogs, etc.) or “minimal risk” activities (*X^2^*(2) = 55.88, *p* < 0.001, *V* = 0.13). Regarding routines, 9.6% of men versus 6.2% women did not maintain a structured daily routine (*X^2^*(2) = 12.03, *p* < 0.01, *V* = 0.06).

Regarding the question of which element concerned them most during the pandemic, significant differences were found with respect to concern about the health of their loved ones (*X^2^*(1) = 4.85, *p* = 0.03, *V* = 0.04) and concern about the psychological impact of dependents (*X^2^*(1) = 6.46, *p* < 0.01, *V* = 0.04). In all these cases, women were more concerned than men were. 

### 3.2. Odds Ratio

Female gender was associated with an augmented risk of intrusive thoughts (*OR* = 2.72 with a CI-95% = 6.64–7.34), avoidance (*OR* = 2.30 with a CI-95% = 7.46–8.04) and hyperactivation (*OR* = 1.75 with a CI-95% = 3.80–4.30). The total risk of presenting symptoms related to posttraumatic symptoms was greater in women than in men (*OR* = 6.77 with a CI-95% = 18.03–19.55). 

Women showed an increased risk of suffering from stress (*OR* = 2.32, CI-95% = 5.01–5.69), depression (*OR* = 0.69, CI-95% = 3.26–3.80), and anxiety (*OR* = 1.58, CI-95% = 2.31–2.84). The total risk of presenting symptoms related to emotional distress was greater in women than in men (*OR* = 4.59, CI-95% = 10.68–12.24).

### 3.3. Univariate General Linear Models by Gender

The model proposed for DASS-21 scores in men are presented in Table 4. A total of 909 people were considered for the model, which explained 21.2% of the variance in DASS-21, and all the variables were significant (see Table 4). With respect to the parameters of the model (see Appendix A, Table A1), men with the highest scores on the DASS-21 present the following characteristics: being young (between 18 and 33), considering to themselves to possibly be at risk of COVID-19, being infected by COVID-19, taking more medication for relaxation or sleep than before, not having any routine during the day, drinking more alcohol than before, having a poor perception of their physical condition, being concerned about the health system overload and having no concern about becoming ill or dying.

The model proposed to explain DASS-21 scores in women (see Table 4) includes data from a total of 2571 people. All variables are significant, explaining 26% of the variance in DASS-21 scores.

Regarding the magnitude and signs of the parameters, women with the highest scores the DASS-21 were young (between 18 and 33 years old), had a low level of education, may be at risk of COVID-19, may have had contact with COVID-19, took medication for relaxation or sleep, did not follow any routines during the day in quarantine, smoke and used more alcohol than before the pandemic, had a negative perception of their physical condition and were concerned about loneliness (see Appendix B, Table A2).

The model proposed for explaining the IES-R scores in men (see Table 5) considered a total of 909 men and explains 13.7% of the variable variance. 

Regarding the magnitude and signs of the parameters (see Appendix C, Table A3), men with the highest scores on the IES-R present the following characteristics: being young (between 18 and 33), not knowing if they are at risk of COVID-19, being infected by COVID-19, taking more medication for relaxation or sleep than before, drank more alcohol than before confinement, being concerned about their economic circumstances and being concerned about loneliness.

Finally, the last model explains the IES-R scores in women, considering a total of 2149 people (see Table 5), and explaining 19% of the variance in IES-R scores. Women with the highest scores on the IES-R were young (between 18 and 33), had a low level of education, may have been at risk of COVID-19, may have been in contact with COVID-19, took more sleep medication than before, did not follow any routine, smoked more than before the pandemic, were concerned about becoming ill or dying, lived with their partner and children or with their parents and had a negative perception of their physical condition (see Appendix D, Table A4).

The model tested first in each of the four previous cases included other nonsignificant variables. The interaction terms between variables were nonsignificant and were removed.

## 4. Discussion

As in the Wang [6], Zhong [15], and Mazza [8] studies, most respondents were women. This gender difference in the sample composition has different possible explanations: (1) women may be more motivated because they are more distressed; (2) men are less responsive because they may tend to avoid coping mechanisms and (3) differences in social media use or disposition to cover questionnaires. The first explanation is consistent with previous studies, which suggest women may be more sensitive and predisposed to depressive and anxiety symptoms [34]. They are also more likely to develop posttraumatic symptoms [35], as shown in our study, which was also consistent with previous literature during this pandemic [11]. Being more distressed and concerned about their mental health may have pushed women to get involved in a study on these characteristics. With respect to the second explanation, some studies have reported that women are more likely to use mental health services [36] and that men have more difficulties seeking help [37] specially after trauma exposure [38]. Other authors have suggested that using an avoidance mechanism in posttraumatic conditions is more frequent in men than in women [39]. During the COVID-19 pandemic, some interesting results must be noted. In the Chinese study by Wang [6], men reported less psychological impact but worse mental status with more symptoms of stress, anxiety and depression. In addition, we cannot eliminate the possibility that other issues linked to the research protocol or the use of social media may have influenced the fact that the majority of participants were women. In this line, a Spanish study on the use of social media pointed that women use them for communication while men use them for consumption and leisure reasons, which could make men less likely to 0get involved in a survey like the one proposed [40].

In our study, more men than women were married, which is considered a protective factor for mental health [39]. Nevertheless, more men lived alone and might have been more isolated during the quarantine, which did not lead to worse psychological responses for men; this finding coincides with various studies revealing how loneliness is less well tolerated in women and in those living alone and without children [41]. During the quarantine, men may be also more exposed to COVID-19 than women because they may be involved in risky work environment, but their perception of being at risk was lower than that reported by women, which might be considered a resilience factor, although not all studies have reported this finding consistently [42]. Zhong et al. [15] described more knowledge of the disease, better prevention attitudes and more optimism in women than in men.

In our sample, women suffered from more anxiety, depression, and stress than men did. Women were also more psychologically impacted and showed more posttraumatic symptoms than men (19% of women vs. 7.4% of men showed severe psychological impacts). The higher prevalence of anxiety, depression and posttraumatic symptoms in women when compared to men has been reported in pre-pandemic studies in many countries included Spain [43] and has been replicated during this pandemic [6,8,11,17]. 

Biological, psychological, and cultural factors are involved in this association. Among them, we highlight that women generally tend to assume a caregiving role in their families, having to balance it with work and, usually household tasks, which makes them more vulnerable in this situation of overload [7]. Other authors thought that these differences are due to increased past trauma exposure in women and not to a gender-specific vulnerability [11]. As women are more likely to suffer traumatic events along their lives, a new trauma as the pandemic is may affect them through a cumulative trauma bias. 

Sleeping difficulties, somatization, hypochondria, and claustrophobia were higher in women, which is consistent with other studies [44]. The use of drugs during quarantine also differed in both genders. It has been reported that females escalate drug use more rapidly than males do, and relapse is more likely to be triggered by stressful events or drug-related cues [45]. Our results are consistent with women using more psychiatric drugs than usual during quarantine and men reporting greater decreases in the use of alcohol and other drugs. Women showed worse general self-care patterns, were involved in less positive activities and engaged in more self-destructive behaviors. An interesting study reported that while men were found to have more than quadruple the risk of poor self-care than women were, men were also found to be approximately 60% more likely to have adequate self-care confidence than women [46]. Therefore, women may have worse self-care behaviors, or they may be reporting them less accurately.

Although some variables used to predict psychological symptoms did not vary by gender, such as age, lack of good routines, intake of medication for relaxation or sleep and consumption of more alcohol than before confinement, other variables were gender specific. In men, being infected by COVID-19 was one of the significant variables predicting IES-R and DASS-21 scores, and the possibility of being at risk was significant only for predicting the impact but not the mental health status measured by the DASS-21. These findings could be related to traditional gender roles in a patriarchal society. It is interesting to note that women use to rate their health status as worse than men (subjectively), even when it is similar (objectively) [47], and that men tend to avoid speaking about their health and that they show a delay when seeking help [48]. On the other hand, the possibility of being at risk was significant in women for both scales, so uncertainty may be more harmful for women than men. Furthermore, uncertainty has been related to worse life quality and anxiety in women suffering from diverse diseases [49]. 

The concerns with predictive value for the IES-R differed by gender. While men’s concerns about their personal finances and loneliness were significantly predictive of their psychological impact, women’s psychological impact was better predicted by concerns about becoming ill or dying. For DASS-21 scores, overload of the health system was the most significant concern for men, while women’s main concern was loneliness.

Secondary and tertiary prevention interventions must consider a gender perspective. Training coping strategies to lead with uncertainty and promoting cooperative models to assume care and housework between men and women, might be particularly important to improve women mental health outcomes. In our study, living with a partner and children was one of the predictors of stronger post-traumatic symptoms for women but not for men. This can be explained either by an overload due to the compatibility of domestic and care tasks with work tasks, or by a greater empathy and concern for the health of the family in women. On the other hand, Government measures securing the economic future of workers and the proper functioning of health systems may especially help men to feel less worry during the pandemic. The evaluation of possible avoidance mechanisms in men must be also considered. From a critical point of view, this does not mean that we should focus our interventions specifically in these concerns (which are influenced by gender stereotypes), but we must consider their origins and consequences in mental health. Finally, it is important to highlight that increasing social support must be useful for both genders, although men tend to ask less for help. Resilience, adaptive coping strategies and social support have been linked to better mental health performing during the pandemic [50]. Psychosocial interventions have been shown to be especially useful in reducing loneliness [51] and anxiety during the pandemic [52], in reducing alcohol consumption [53], and even improving physical health [54]. Finally, the fact of maintaining and healthy daily routines, as well as avoiding alcohol consumption are useful general recommendations in both genders as these behaviors showed a protective effect on mental health.

In summary, women showed worse psychological responses and should be considered a vulnerable group, but men were underrepresented in the sample. Of course, for the generalization of the results, the limitations the current situation of confinement and the risk of contagion have imposed, such as the impossibility of carrying out a randomized study and clinical interviews or the need to apply an online survey, which must be taken into account. Additionally, results are not generalizable to individuals identifying as nonbinary (gender variants other than male and female).

## 5. Conclusions

Despite our study overrepresenting women with master-level education, it seems like women as a population can be considered a particularly vulnerable group in this pandemic. This vulnerability can be seen in relation to the development of a wide range of symptomatology which scopes from mild to moderate; such as psychological and somatic anxiety, depression, stress and symptoms in the posttraumatic area. Biological, psychological, and social factors may be involved in these gender differences, and finally, future psychosocial interventions must consider the gender perspective.

## Figures and Tables

**Figure 1 ijerph-18-03731-f001:**
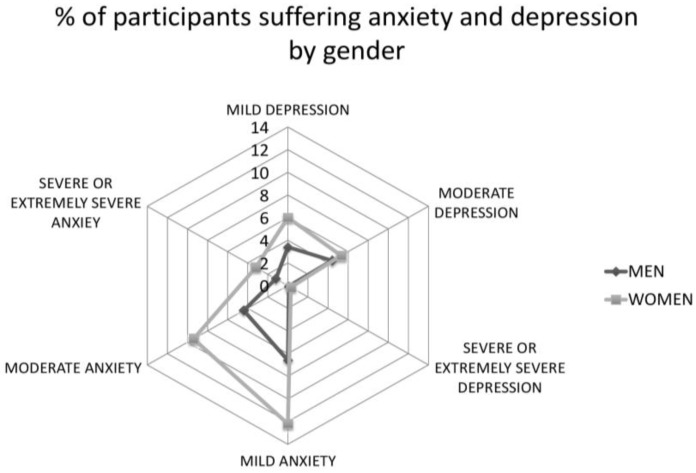
Percentage of participants suffering anxiety and depression by gender.

**Table 1 ijerph-18-03731-t001:** Gender differences in sociodemographic variables.

Demographic Variable	*X^2^*(df)	*p*	*V*	Subgroup		Men (*n* = 909)	Women (*n* = 2611)	Total (*n* = 3250)
Age	61.63(4)	<0.001	0.13	18–25 years	*n*	105	352	457
					%	11.6%	13.5%	13%
				26–33 years	*n*	155	573	728
					%	17.1% *	21.9% *	20.7%
				34–45 years	*n*	346	1037	1383
					%	38.1%	39.7%	39.3%
				46–60 years	*n*	206	542	748
					%	22.7%	20.8%	21.3%
				>61 years	*n*	97	107	204
					%	10.7% *	4.1% *	5.8%
Marital status	17.51(4)	0.00	0.07	Single	*n*	225	627	852
					%	24.9%	24.1%	24.3%
				In partnership	*n*	240	818	1058
					%	26.6% *	31.5% *	30.2%
				Married	*n*	391	963	1354
					%	43.3% *	37.1% *	38.7%
				Separated/ Divorced	*n*	37	161	198
					%	4.1% *	6.2% *	5.7%
				Widow/er	*n*	9	28	37
					%	1%	1.1%	1.1%
Educational level	18.88(2)	<0.001	0.07	Undergraduate	*n*	255	549	804
					%	28.3% *	21.4% *	23.2%
				Graduate	*n*	351	1060	1411
					%	39%	41.2%	40.7%
				Postgraduate	*n*	294	962	1256
					%	32.7% *	37.4% *	36.2%
Job occupation	82.44(8)	<0.001	0.17	Farmer, rancher or similar	*n*	6	7	13
					%	0.8%	0.3%	0.4%
				Entrepreneurs	*n*	51	92	143
					%	6.4% *	4.2% *	4.7%
				Elementary occupations	*n*	24	97	121
					%	3%	4.4%	4%
				Official	*n*	37	20	57
					%	4.6% *	0.9% *	1.9%
				Facilites	*n*	6	2	8
					%	0.8% *	0.1% *	0.3%
				Administrat	*n*	48	185	233
					%	6.0% *	8.4% *	7.7%
				Professional	*n*	396	1249	1645
					%	49.5% *	56.4% *	54.6%
				Technical	*n*	163	345	508
					%	20.4% *	15.6% *	16.8%
				Assistant	*n*	69	218	287
					%	8.6%	9.8%	9.5%
Family cohabitation	15.36(5)	0.01	0.07	Alone	*n*	120	246	366
					%	14.5% *	11.3% *	12.1%
				Partner	*n*	251	596	847
					%	30.3% *	27.3% *	28.1%
				Partner and children	*n*	291	821	1112
					%	35.1%	37.6%	36.9%
				Parents	*n*	113	407	540
					%	35.1%	37.6%	36.9%
				Parents-in-law	*n*	3	2	5
					%	0.4%	0.1%	0.2%
				Partner, children and other relatives	*n*	31	112	143
					%	3.7%	5.1%	4.7%
Children at home	14.02(4)	0.01	0.06	I do not have chindren	*n*	463	1305	1768
					%	51%	50.6%	50.7%
				None	*n*	98	193	291
					%	10.8% *	7.5% *	8.3%
				One	*n*	167	492	659
					%	18.4%	19.1%	18.9%
				Two	*n*	140	490	630
					%	15.4% *	19.0% *	18.1%
				Three or more	*n*	39	101	140
					%	4.3%	3.9%	4%
Time at home	55.88(2)	<0.001	0.13	I’ve been home all day	*n*	399	1480	1879
					%	45.3% *	59.8% *	56.0%
				I’ve come out just for the essentials	*n*	469	967	1436
					%	53.2% *	39.1% *	42.8%
				I’ve come out	*n*	13	26	39
					%	1.5%	1.1%	1.2%

Notes: Asterisk = residual ± 1.97; *X^2^* = Chi Squared test; *df* = degree of freedom; *p* = significance; *V* = Cramer’s V test.

**Table 2 ijerph-18-03731-t002:** Association between gender and mental health state during the pandemic (*n* = 3520).

Variable	Group	*n*	*M*	*SD*	*t*	*p*	*d*
General selfcare	Men	909	3.02	0.99	−2.14	0.033	0.08
	Women	2611	3.10	1.08			
No positive activities	Men	909	2.47	1.20	−2.19	0.028	0.08
	Women	2611	2.57	1.31			
Self-destructive behavior	Men	909	2.65	1.22	−4.96	<0.001	0.19
	Women	2611	2.90	1.34			
Global Impact of Event	Men	909	18.80	11.29	15.05	<0.001	0.58
	Women	2611	25.56	11.81			
Intrusion	Men	909	6.70	4.90	−4.00	<0.001	0.58
	Women	2611	9.71	5.48			
Avoidance	Men	909	7.75	4.61	13.23	<0.001	0.51
	Women	2611	10.05	4.47			
Hyperactivation	Men	909	4.05	3.40	12.78	<0.001	0.47
	Women	2611	5.80	4.00			
Global Mental health	Men	909	11.46	10.78	−0.62	<0.001	0.39
	Women	2611	16.05	12.42			
Stress	Men	909	5.35	4.68	12.41	<0.001	0.46
	Women	2611	7.67	5.33			
Depression	Men	909	3.53	3.85	−4.53	<0.001	0.17
	Women	2611	4.22	4.21			
Anxiety	Men	909	2.58	3.41	11.21	<0.001	0.41
	Women	2611	4.16	4.30			
Emotional state	Men	904	7.06	1.6	8.55	<0.001	0.19
	Women	2569	6.52	1.7			

Notes: n = sample size; *M* = Mean; *SD* = Standard Deviation; *t* = T-Test; *p* = significance; *d* = Cohen’s d.

**Table 3 ijerph-18-03731-t003:** Gender differences on behavioral variables.

Variable	*X^2^*(*df*)	*p*	*V*	Subgroup		Men (*n* = 909)	Women (*n* = 2611)	Total (*n* = 3520)
Medication	25.05(3)	<0.001	0.08	Equal	*n*	78	306	384
					%	8.6% *	11.7% *	10.9%
				More than usual	*n*	20	143	163
					%	2.2% *	5.5% *	4.6%
				Less than usual	*n*	9	26	35
					%	1%	1%	1%
				No medication	*n*	802	2136	2938
					%	88.2% *	81.8% *	83.5%
Alcohol	110.68(3)	<0.001	0.18	Equal	*n*	315	670	985
					%	34.7% *	25.7% *	28.0%
				More than usual	*n*	74	195	269
					%	8.1%	7.5%	7.6%
				Less than usual	*n*	213	360	573
					%	23.4% *	13.8% *	16.3%
				I do not drink	*n*	307	1386	1693
					%	33.8% *	53.1% *	48.1%
Drugs	31.41(3)	<0.001	0.09	Equal	*n*	35	42	77
					%	3.9%	1.6%	2.2%
				More than usual	*n*	13	24	37
					%	1.4%	0.9%	1.1%
				Less than usual	*n*	18	16	34
					%	2.0% *	0.6% *	1.0%
				I do not take drugs	*n*	843	2529	3372
					%	92.7% *	96.9% *	95.8%
Sleep problems	47.16(3)	<0.001	0.10	Equal	*n*	229	679	908
					%	25.2%	26%	25.8%
				More than usual	*n*	132	644	776
					%	14.5% *	24.7% *	22.0%
				Less than usual	*n*	30	70	100
					%	3.3% *	2.7% *	2.8%
				I do not have	*n*	518	1218	1736
					%	57.0% *	46.6% *	49.3%
Somatizations	127.15(3)	<0.001	0.19	Equal	*n*	201	801	1002
					%	22.1% *	30.7% *	28.5%
				More than usual	*n*	141	707	848
					%	15.5% *	27.1% *	24.1%
				Less than usual	*n*	12	66	78
					%	1.3% *	2.5% *	2.2%
				I do not feel it	*n*	555	1037	1592
					%	61.1% *	39.7% *	45.2%
Claustrophobia	43.50(3)	<0.001	0.11	Equal	*n*	82	371	453
					%	9.0% *	14.2% *	12.9%
				More than usual	*n*	64	303	367
					%	7.0% *	11.6% *	10.4%
				Less than usual	*n*	3	33	36
					%	0.3% *	1.3% *	1.0%
				I do not feel it	*n*	760	1904	2664
					%	83.6% *	72.9% *	75.7%
Hypocondria	16.73(3)	<0.001	0.07	Equal	*n*	157	486	643
					%	17.3%	18.6%	18.3%
				More than usual	*n*	88	381	469
					%	9.7% *	14.6% *	13.3%
				Less than usual	*n*	10	27	37
					%	1.1%	1%	1.1%
				I do not have	*n*	654	1717	2371
					%	71.9% *	65.8% *	67.4%

Notes: Asterisk = residual ±1.97; *X^2^* = Chi Squared test; *df* = degree of freedom; *p* = significance; *V* = Cramer’s V test.

**Table 4 ijerph-18-03731-t004:** Univariate general linear model for predicting DASS-21 in men and women.

	*SS*	*Df*	*MS*	*F*	*p*	*ŋ^2^_p_*
Men						
Corrected model	24,138.10	19	1270.42	13.86	<0.001	0.23
Intersection	18,137.99	1	18,137.99	197.93	<0.001	0.18
Age	3210.66	3	1070.22	11.67	<0.001	0.04
At risk for COVID-19	3734.66	2	1867.33	20.37	<0.001	0.04
Infected by COVID-19	1320.12	2	660.06	7.20	0.001	0.02
Relaxing/Sleep medication	6137.16	3	2045.72	22.32	<0.001	0.07
Routine	1263.76	2	631.88	6.89	0.001	0.02
Alcohol use	740.50	3	246.83	2.69	0.045	0.01
Physical condition	837.19	2	418.59	4.56	0.011	0.01
Concerns about becoming ill/death	586.02	1	586.02	6.39	0.012	0.01
Concerns about health system overhead	1605.94	1	1605.94	17.52	<0.001	0.02
Error	81,463.59	889	91.63			
Total	224,956.00	909				
Corrected Total	105,601.70	908				
Women						
Corrected model	103,724.22	25	4148.96	36.20	<0.001	0.26
Intersection	92,667.18	1	92,667.18	808.55	<0.001	0.24
Age	6739.66	3	2246.55	19.60	<0.001	0.02
Education	2908.14	4	727.03	6.34	<0.001	0.01
At risk for COVID-19	5495.15	2	2747.58	23.97	<0.001	0.02
Relaxing/Sleep medication	30,730.02	3	10,243.34	89.37	<0.001	0.09
Routine	8757.41	2	4378.70	38.20	<0.001	0.03
Smoke	2278.16	3	759.38	6.62	<0.001	0.01
Alcohol use	2691.66	3	897.22	7.82	<0.001	0.01
Physical condition	13,113.55	2	6556.77	57.21	<0.001	0.04
Concerns about loneliness	2286.92	1	2286.92	19.95	<0.001	0.01
Contact with infected by COVID-19	811.21	2	405.60	3.53	0.029	0.00
Error	291,676.76	2545	114.60			
Total	1,053,481.00	2571				
Corrected Total	395,400.99	2570				

Notes: *SS* = Sum of Square; *Df* = Degree of freedom; *MS* = Mean Square; *F* = F-test; *ŋ^2^_p_* = partial eta-squared, considering reference values of 0.01, 0.06 and >0.14 as small, medium and large sizes respectively [34]. For men, R-square = 0.23 (adjusted R-square = 0.21). For women R-square = 0.26 (adjusted R-square = 0.26).

**Table 5 ijerph-18-03731-t005:** Univariate general linear model for predicting IES in men and women.

	*SS*	*Df*	*MS*	*F*	*p*	*ŋ^2^_p_*
Men						
Corrected model	17,529.18	15	1168.61	10.61	<0.001	0.15
Intersection	31,805.14	1	31,805.14	288.87	<0.001	0.24
Age	2872.52	3	957.50	8.69	<0.001	0.03
At risk for COVID-19	3186.44	2	1593.22	14.47	<0.001	0.03
Infected by COVID-19	816.25	2	408.12	3.70	0.025	0.01
Relaxing/Sleep medication	3946.80	3	1315.60	11.94	<0.001	0.04
Alcohol use	2053.24	3	684.41	6.21	<0.001	0.02
Concerns about own economy	470.35	1	470.35	4.27	0.039	0.01
Concerns about loneliness	798.26	1	798.26	7.25	0.007	0.01
Error	98,317.93	893	110.09			
Total	436,891.00	909				
Corrected Total	115,847.11	908				
Women						
Corrected model	59,283.69	29	2044.26	18.02	<0.001	0.19
Intersection	108,942.85	1	108,942.85	960.38	<0.001	0.31
Age	4312.68	3	1437.56	12.67	<0.001	0.02
Education	3772.37	4	943.09	8.31	<0.001	0.02
At risk for COVID-19	3895.21	2	1947.60	17.16	<0.001	0.02
Contact with COVID-19	919.56	2	459.78	4.05	0.018	0.01
Relaxing/Sleep medication	16,408.34	3	5469.44	48.21	<0.001	0.06
Routine	2474.48	2	1237.24	10.90	<0.001	0.01
Smoke	1279.48	3	426.49	3.76	0.010	0.01
Alcohol use	1905.01	3	635.00	5.59	0.001	0.01
Concerns about becoming ill/death	2212.16	1	2212.16	19.50	<0.001	0.01
Living together	1808.26	4	452.06	3.98	0.003	0.01
Physical condition	5531.58	2	2765.79	24.38	<0.001	0.02
Error	240,372.59	2119	113.43			
Total	1,684,348.00	2149				
Corrected Total	299,656.28	2148				

Notes: *SS* = Sum of Square; *Df* = Degree of freedom; *MS* = Mean Square; *F* = F-test; *ŋ^2^_p_* = partial eta-squared. considering reference values of 0.01, 0.06 and >0.14 as small. medium and large sizes respectively [34]. For men, R-square = 0.15 (adjusted R-square = 0.14), while for women R-square = 0.20 (adjusted R-square = 0.19).

## Data Availability

Data supporting reported results can be obtained by contacting the authors.

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
