# Peer review of "Psychological Responses According to Gender during the Early Stage of COVID-19 in Spain"

_ijerph, 2021, doi:10.3390/ijerph18073731_

Round 1
Reviewer 1 Report
The psychological responses to the COVID-19 is a very important topic. This study focuses on gender differences in mental health in the period of emergency of COVID-19. To further improve the qualtiy of this study, I put forward the following questions and comments for this research:
- It is better to prove that women are more likely to get bad mental health than men is not because the number of sampled female respondents is more than that of male respondents. I think the authors have recognized this problem. But the more detailed explanation and discussion are needed.
-
The limitations of network questionnaire survey need to be explained. especially, how does that affect the outcomes?
- Do the different roles that different genders play in families also affect their mental health?
- It will be better to give professional suggestions and policy recommendations on how to response the mental health problems facing COVID-19.
Author Response
Thank you very much for your comments and suggestions. We respond below.
1. It is better to prove that women are more likely to get bad mental health than men is not because the number of sampled female respondents is more than that of male respondents. I think the authors have recognized this problem. But the more detailed explanation and discussion are needed.
Response:
Thank you for your comment. The sample composition concerning gender has been discussed around three hypotheses: 1) women may be more motivated to respond because they are more distressed; 2) men are less responsive because they may tend to avoid coping mechanisms and 3) differences in social media use or disposition to cover questionnaires.
Further detail has been considered for hypothesis 1 and 2. The third hypothesis was insufficiently argued, so we have completed it and included a new reference (lines 326-330)
2. The limitations of network questionnaire survey need to be explained. especially, how does that affect the outcomes?
Response:
The use of an online survey is a limitation that we share with other research conducted in the very early stages of the pandemic. In any case, some precautions were considered. The survey was linked to an email address, so participants could not access the platform twice. In order to avoid random answering patterns, we checked carefully the data base, analyzing patterns of responses, possible missing data, outliers and time to answer the survey. We decided to delete 44 abnormal responses. This issue has been included in the manuscript (lines 165-171)
3. Do the different roles that different genders play in families also affect their mental health?
Response:
We have completed this issue including this paragraph:
“The higher prevalence of anxiety, depression and posttraumatic symptoms in women when compared to men has been reported in prepandemic studies in many countries, included Spain [32]; which has been replicated during this pandemic [4][5][8][8][10].
Biological, psychological and cultural factors are involved in this association. Among them, we highlight that women generally tend to assume a caregiving role in their families, having to balance it with work and, usually household tasks, which makes them more vulnerable in this situation of overload [13]. Other authors thought that these differences are due to increased past trauma exposure in women and not to a gender-specific vulnerability [4].” (lines 345-354)
4. It will be better to give professional suggestions and policy recommendations on how to response the mental health problems facing COVID-19.
Response:
We have completed this issue based on gender differences because general recommendations have been widely published before.
We have included this paragraph (lines 386-406):
“Secondary and tertiary prevention interventions must consider a gender perspective. Training coping strategies to lead with uncertainty and promoting cooperative models to assume care and housework between men and women, might be particularly important to improve women mental health outcomes. In our study, living with a partner and children was one of the predictors of stronger post-traumatic symptoms for women but not for men. This can be explained either by an overload due to the compatibility of domestic and care tasks with work tasks, or by a greater empathy and concern for the health of the family in women. On the other hand, Government measures securing the economic future of workers and the proper functioning of health systems may especially help men to feel less worry during the pandemic. The evaluation of possible avoidance mechanisms in men must be also considered. From a critical point of view, this doesn´t mean that we should focus our interventions specifically in these concerns (which are influenced by gender stereotypes), but we must consider their origins and consequences in mental health. Finally, it is important to highlight that increasing social support must be useful for both genders, although men tend to ask less for help. Resilience, adaptive coping strategies and social support have been linked to better mental health performing during the pandemic [51]. Psychosocial interventions have been shown to be especially useful in reducing loneliness[52] and anxiety during the pandemic [53], in reducing alcohol consumption [54] and even improving physical health [55] . Finally, the fact of maintaining and healthy daily routines, as well as avoiding alcohol consumption are useful general recommendations in both genders as these behaviors showed a protective effect on mental health.”
Reviewer 2 Report
Río-Casanova et al examined whether there were gender differences in mental health symptoms associated with the pandemic in Spain. This study is rich in data and examines a variety of variables associated with mental health, protective factors, quality of life, etc. Some concerns remain.
- The importance of this study is not highlighted in the manuscript. What is the value of understanding whether there are gender differences in mental health symptoms, particularly less than a month after the declaration of the state of emergency?
- The authors only report the percentage of household characteristics for 71% of the sample. Please clarify the household characteristics of the other 29%. Similarly, only 88.4% is accounted for in housing.
- There is a significant gap in ages (34-60). Is this because there were no significant differences for the other groups? If so, I strongly suggest adding those values even if not significant.
- It is unclear when Games-Howell and Tukey test were used. Similarly, please discuss how the imbalance of sample size for men and women was considered in the analyses, particularly since unequal variances between samples may affect the assumption of equal variances in tests like ANOVAs.
- Please report whether comparisons survived multiple comparisons.
- I’m curious about how many individuals lost their jobs or were not working during that time as this could significantly contribute to anxiety, etc.
- It is interesting and unexpected that men without concern about becoming ill or dying had higher scores in the DSS-21 (and the opposite for women). Why may this be the case?
- It would be interesting to examine the role of geographic location (rural vs urban).
- While the article focuses on gender differences, it is also very interesting that mental health seems to vary by age. More attention may be given to this in the manuscript.
- The authors may consider highlighting more the role of protective factors in the mental health of these individuals, as this can have significant implications for how mental health is managed.
Author Response
Thank you very much for your comments and suggestions. We responde below.
1. The importance of this study is not highlighted in the manuscript. What is the value of understanding whether there are gender differences in mental health symptoms, particularly less than a month after the declaration of the state of emergency?
Response:
Thank you for your comment. The importance of the study has been made explicit in the text with this paragraph:
“Finally, the evolution in the population mental health along the pandemic is also an important issue. Some longitudinal studies have been recently published and confirm the deterioration in the population mental health comparing pre-pandemic to pandemic data [21][22][23]. Other two longitudinal studies have shown that this deterioration persisted in several measures in the following months and that gender differences remained [24][25]. There is a lack of longitudinal research focused on gender along the pandemic. Taking this into account, the analysis of men and women mental health status at the early stages of the pandemic, could be useful in order to propose gender adjusted interventions that may decrease the impact of the pandemic and have positive effects along time. “ (lines 84-93).
2. The authors only report the percentage of household characteristics for 71% of the sample. Please clarify the household characteristics of the other 29%. Similarly, only 88.4% is accounted for in housing.
Response:
Thank you for clarifying this point. We have added the percentages to complete the information.
3. There is a significant gap in ages (34-60). Is this because there were no significant differences for the other groups? If so, I strongly suggest adding those values even if not significant.
Response:
Because of space requirements, we decided to include only the most relevant information in the Tables. We agree with the reviewer that is important to report both significant and non-significance differences between groups, so we are grateful to include this information in Table 1 and 3.
On the other hand, we included all age ranges in the univariate general linear model (18-33, 34-49, 50-65 and 66-79), as is reported in Appendix A-D. Possibly this is misunderstood because in the result section we have reported the more important information but, indeed, when comparing with participants aged from 66 to 79, those participants men aged from 18 to 33 and from 34 to 49 had higher scores in DASS-21 (B = 6.06, p <.001; B = 4.05, p = .007, respectively) and in IES (B = 5.49, p = .001; B = 4.23, p = . .010, respectively). The comparison between participants aged from 66 to 79 and those from 50 to 65 did not resulted statistically significant. In women we replicated the same result, when comparing with participants aged from 66 to 79, those women from 18 to 33 and from 34 to 49 reported higher scores in DASS-21 (B = 6.76, p <.001; B = 4.87, p = .009, respectively) and in IES (B = 7.33, p <.001; B = 5.59, p = .005, respectively).
4. It is unclear when Games-Howell and Tukey test were used. Similarly, please discuss how the imbalance of sample size for men and women was considered in the analyses, particularly since unequal variances between samples may affect the assumption of equal variances in tests like ANOVAs.
Response:
We had a mistake in the manuscript. We described that we used T-Test or ANOVA but we only used T-test because all the comparison was between two groups (men and women). We did not used ANOVA with the correction for multiple comparison Games-Howell or Tukey. Instead of that we applied it in the univariate general linear model.
We have modified it in the manuscript; concretely the Statistical Analysis section:
“We compared one quantitative and one categorical variable (with two groups) through Student’s t-tests. We used the robust Welch’s Test whether the assumption of homo-geneity of variances (Levene’s Test) was violated. We used Chi-square test with the McNemar test correction for 2x2 tables and the Bonferroni correction for multiple test-ing.”
“We analyzed the homogeneity variance assumption, and we considered the correction of Bonferroni or Games-Howell for multiple testing (depending on the homogeneity variance assumption was accepted or violated).” (lines 152-154).
Respect to the imbalance of sample size for men and women, we recognize the possibility to have unequal variances between groups and we developed the process previously explained. That is, we used robust Welch’s Test when the assumption of equal variances between groups was violated. Further, we always report the effect size to really advocate the magnitude of the difference between groups. Depends on the analysis, we reported the appropriate statistic: Cramer’s V for Chi Square and Cohen’s d for ANOVA or T-test.
5. Please report whether comparisons survived multiple comparisons.
Response:
Thank you very much for your comment. This is something that we had not considered previously for T-test, and we have included in this revision.
When developing T-test, one common method of adjustment for multiple testing is Bonferroni. Because we developed twelve tests (see Table 2), the p value than we must use for rejecting the null hypothesis is 0.00416 (.05/12). We have clarified this in the manuscript. Concretely, we have included the following paragraph:
“For multiple T-test we apply the Bonferroni correction, considering a level of significance of .00416 for rejecting the null hypothesis.” (lines 158-160).
Accordingly, we have modified the results, as you can see in the following paragraph: “Women reported higher scores than men, but non-significantly difference, regarding self-care patterns (t(1) = -2.14, p = .03 > p =.00416, d = 0.08) and no positive activities subscale (t(1) = -2.19, p = .03 > p =.00416, d = 0.08). On the other hand, significantly differences between groups were found on self-destructive behaviors subscale (t(1) = -4.96, p < .001, d = 0.19).”
The other comparison analysis, chi-square, and univariate general linear model, we directly included the correction for multiple testing. We have specified in the manuscript, as we showed previously in the comment 4.
6. I’m curious about how many individuals lost their jobs or were not working during that time as this could significantly contribute to anxiety, etc.
Response:
We agree that this is something very interesting to analyze. Recent job loss was not surveyed as at the moment, because the emergency state had been recently declared and the future for many jobs was unclear. At that time, we considered that knowing the occupation and work sector was more important. These results are summarized in Table 1.
Nevertheless, we agree with the reviewer that this issue is of particular interest for future research (specially longitudinal and prospective studies) and we will consider the suggestion in the longitudinal study our research group is conducting.
7. It is interesting and unexpected that men without concern about becoming ill or dying had higher scores in the DSS-21 (and the opposite for women). Why may this be the case?
Response:
We have included this question in the discussion. The concerns about health systems and government measures specially in the field of work policies and economics correlated with worse mental health state in men, while women were more worried about their health and the health of the people they love. It could be related to gender roles in a patriarchal society. We have included the following paragraph:
“These findings could be related to traditional gender roles in a patriarchal society. It is interesting to note that women use to rate their health status as worse than men (subjectively), even when it is similar (objectively) [48], and that men tend to avoid speaking about their health and that they show a delay when seeking help [49]. “(lines 372-375)
This issue must be considered to include a gender perspective to the psychosocial interventions addressed to the general population during the pandemic, as we propose in the following paragraph:
“Secondary and tertiary prevention interventions must consider a gender perspective. Training coping strategies to lead with uncertainty and promoting cooperative models to assume care and housework between men and women, might be particularly important to improve women mental health outcomes. In our study, living with a partner and children was one of the predictors of stronger post-traumatic symptoms for women but not for men. This can be explained either by an overload due to the compatibility of domestic and care tasks with work tasks, or by a greater empathy and concern for the health of the family in women.
On the other hand, Government measures securing the economic future of workers and the proper functioning of health systems may especially help men to feel less worry during the pandemic. The evaluation of possible avoidance mechanisms in men must be also considered. From a critical point of view, this doesn´t mean that we should focus our interventions specifically in these concerns (which are influenced by gender stereotypes), but we must consider their origins and consequences in mental health. Finally, it is important to highlight that increasing social support must be useful for both genders, although men tend to ask less for help. Resilience, adaptive coping strategies and social support have been linked to better mental health performing during the pandemic [49]. Psychosocial interventions have been shown to be especially useful in reducing loneliness[50] and anxiety during the pandemic [51], in reducing alcohol consumption [52] and even improving physical health [53] . “ (lines 386-407).
8. It would be interesting to examine the role of geographic location (rural vs urban).
Response:
Rural vs urban location has not been evaluated, only bigger geographical areas (region) have been surveyed. We consider this suggestion very interesting but as it wasn´t the aim of the study, it wasn´t survey and therefore cannot be analyzed.
9. While the article focuses on gender differences, it is also very interesting that mental health seems to vary by age. More attention may be given to this in the manuscript.
Response:
Indeed, mental health performing during the pandemic is different in different age groups. In our sample, young participants showed worse mental health results (as it is noted in the introduction, this result has been replicated). A vulnerability for both extremes of life has been reported in previous studies. Our research group has already published a paper studying the present sample focusing on age differences. For the present paper, as there were no age differences between both genders, we focused in other variables which performed different concerning gender.
10. The authors may consider highlighting more the role of protective factors in the mental health of these individuals, as this can have significant implications for how mental health is managed.
Response:
We appreciate your suggestion, which is very interesting, and we have included it in the manuscript.
We have not analyzed protective factors, but we could assume it as the oppositive of risk factor. In that way, for men the more important protective factors (for DASS-21 and IES) would be to be older, to have the self-perception of a good physical or medical condition and without risk of COVID-19, was not infected by COVID-19, taking less or equal medication for relaxation or sleep than before, drinking less or equal alcohol than before and having any routine during the day.
For women, the more important protective factors of mental health would be to be older, a higher level of education, consider that they would be not at risk of COVID-19 or had no contact with the virus, not take medication for relaxation or sleep, follow any routines during the day, smoke and drink less or equal alcohol than before the pandemic and have a good perception of their physical condition.
We included in the text the statement: “Finally, the fact of maintaining healthy daily routines, as well as avoiding alcohol consumption are useful general recommendations in both genders as these behaviors showed a protective effect on mental health.” (lines 404-407)
Reviewer 3 Report
The authors took up a very topical and socially important issue in the context of the SARS-COV-2 pandemic.
However, the conducted research and the presented results leave a certain degree of insufficiency.
It would be desirable to present the results of the research, rather its next stage in terms of recognized scales of personal values, the level of life satisfaction and coherence.
It would be advisable to undertake such a research effort, especially after the lapse of subsequent months, in order to capture the changes taking place in social reactions.
Author Response
The authors took up a very topical and socially important issue in the context of the SARS-COV-2 pandemic.
However, the conducted research and the presented results leave a certain degree of insufficiency.
It would be desirable to present the results of the research, rather its next stage in terms of recognized scales of personal values, the level of life satisfaction and coherence.
It would be advisable to undertake such a research effort, especially after the lapse of subsequent months, in order to capture the changes taking place in social reactions.
We appreciate your comments and suggestions. We responde below.
As the research protocol indicates, the data were collected in the first weeks of the first wave of the pandemic. This made the psychometric instruments used focused on the impact (through the measurement of post-traumatic symptoms) and on stress, anxiety and depression as measures of mental health. The aim was to be able to compare results with other studies carried out at the same time, with which it shares methodology and instruments, in order to be able to assess whether the results differed between countries. Quality of life indicators and satisfaction were not considered a priority at that time, when psychopathological performing prevailed. We agree that nowadays those variables are extremely important, and we will take into account your suggestion for future research in this field.
On the other hand, our research team is performing a longitudinal analysis of this sample because we agree with the importance of knowing how the mental health has evolved across time. The study is being performed and, shortly, we will be able to provide more detail. Nevertheless, we have included some references of longitudinal studies which point out that the mental health deterioration observed at the early stages persisted months after the outbreak. Concretely, we have included lines 84-939: Finally, the evolution in the population mental health along the pandemic is also an important issue. Some longitudinal studies have been recently published and confirm the deterioration in the population mental health comparing pre-pandemic to pandemic data [21][22][23]. Other two longitudinal studies have shown that this deterioration persisted in several measures in the following months and that gender differences remained [24][25]. There is a lack of longitudinal research focused on gender along the pandemic. Taking this into account, the analysis of men and women mental health status at the early stages of the pandemic, could be useful in order to propose gender adjusted interventions that may decrease the impact of the pandemic and have positive effects along time.
We are aware that with regard to Covid19, the studies may seem outdated within a few months of their completion and perhaps this is a limitation of the study. However, its strength lies in its focus on the gender issue, which seems to be forgotten in the literature on mental health in this pandemic.
Reviewer 4 Report
Thanks to allow me the evaluation of the article entitled Psychological Responses According to Gender During Stage of Covid 19 in Spain by Lucia del Rio et al.
Abstract: OK. The Background could be improved. The background should provide a historic framework of the study, Where is the aim?
Keywords: OK
Introduction: The introduction is correct. However, is too short for providing a correct framework for this research.
If no research has been conducted in the general population of occidental countries examining specifically fender differences responses to COVID 19, in this case, the introduction should describe different research carried out and why gender is not taken into consideration. Please provide in the introduction or discussion references in addition to research conducted into your country.
Please clarify the aim. We aimed xxxx....... if you write “ we are particularly interested” we don´t know if it is one aim, one like one preference. Please identify the aim.
Material and Methods.
- Procedures and Instruments: OK. But, please specify limitations How do you know one survey is not done twice?
- There is not IP filtering?
- There are not cookies filtering?
- There is not time filtering?
Statistical Analysis is OK
Results :
All individuals completed the survey? Anybody was excluded? Please indicate the number of people who were excluded from incomplete the survey. Only 4?
Tables: Please check the last valor’s. N should represent 100 % or you indicate the last valor’s of your results or are not coincidence with N and with the percentage. Please Check.
Discussion.
Should provide more information about new evidence and studies.
Author Contributions. With Initials.
Author Response
We are grateful for your comments and suggestions. Thank you very much. We respond below to each one.
1. Abstract: OK. The Background could be improved. The background should provide a historic framework of the study, Where is the aim?
Response:
We have rewritten the Background of the Abstract. Because of the space limitation we decided to include the following paragraph (from line 16 to 18):
“Current research has pointed out an increased risk for mental health problems during the COVID-19 pandemic in women when compared to men, but the reason for these differences remains unclear. The aim of this research is to study early psychological responses to the pandemic in the Spanish general population, focusing on gender differences.”
Keywords: OK
2. Introduction: The introduction is correct. However, is too short for providing a correct framework for this research.
Response:
The introduction has been rewritten. New references have been included, particularly centered in gender and longitudinal studies. The state of knowledge up to date has also been clarified (see from line 40 to line 92).
3. If no research has been conducted in the general population of occidental countries examining specifically fender differences responses to COVID 19, in this case, the introduction should describe different research carried out and why gender is not taken into consideration. Please provide in the introduction or discussion references in addition to research conducted into your country.
Response:
Gender has been analyzed in multiple studies in many countries (including occidental countries and Spain in particular), but it has also been considered one more variable. The available data include only differences in the percentage of psychological symptoms among men and women, with a statistically significant gender difference. Nevertheless, no risk estimations have been conducted and no regression models have been proposed. The relationship between different concerns in men and women manifest with their mental health outcomes has also remained unstudied.
As the gender issue has been marginally addressed and it has not been considered the aim of other studies, we do not have any explanatory models in this regard and there is also a lack of proposal for gender adjusted psychological intervention.
4. Please clarify the aim. We aimed xxxx....... if you write “ we are particularly interested” we don´t know if it is one aim, one like one preference. Please identify the aim.
Response:
The aim of the study has been clarified. Thank you.
In the Abtract: The aim of this research is to study early psychological responses to the pandemic in the Spanish general population, focusing on gender differences.
In the main text (lines 80-83): the aim of this research is to study the role of gender on mental health responses during the early days of the COVID-19 pandemic in the Spanish general population. Based on the results, specific recommendations are postulated.
5. Material and Methods.
- Procedures and Instruments: OK. But, please specify limitations How do you know one survey is not done twice?
- There is not IP filtering?
- There are not cookies filtering?
- There is not time filtering?
Response:
Thank you for you comment. We specify limitations from line 398 to 403. It is true that to apply an online survey present risk, but we tried to manage it, especially because we have no other alternatives in than moment of the pandemic. In this way, the survey was linked to an email address, so participants could not access the platform twice. In order to avoid random answering patterns, we checked carefully the data base, analyzing patterns of responses, possible missing data and outliers and response time. Responding to your questions, we did not apply IP filtering nor cookies filtering, but we excluded participants with the same response in all the questionnaire and those which finished in an impossible time. Finally, we excluded 44 participants because of their “abnormal responses”. We have included this information from line 165 to 171.
Statistical Analysis is OK
6. Results :
All individuals completed the survey? Anybody was excluded? Please indicate the number of people who were excluded from incomplete the survey. Only 4?
Tables: Please check the last valor’s. N should represent 100 % or you indicate the last valor’s of your results or are not coincidence with N and with the percentage. Please Check.
Response:
Thank you for your comments; it is something that we can specify better. Concretely, during the recruitment period, 4139 people completed the survey. It was mandatory to respond all the items to finish and send the responses. We removed 120 subjects because they did not provide informed consent, 21 people were under 18 years old, 430 were respondents from other countries, 44 abnormal responses and 4 participants perceived themselves to be included in other gender categories.
We have included this information in the manuscript (line 168-171):
“During the recruitment period, 4139 people completed the survey. We removed 120 subjects because they did not provide informed consent, 21 people were under 18 years old, 430 were respondents from other countries, 44 abnormal responses and 4 participants perceived themselves to be included in other gender categories (the small number of participants impeded our ability to analyze this specific subgroup). A final sample of 3520 was included.”
On the other hand, we have revised all summations. Possibly, the misunderstood was that, because of the space requirements, we only reported selected categories, which makes that the sum was not the 100%. We have modified it including all categories and now all the sums are correct.
7. Discussion.
Should provide more information about new evidence and studies.
Response:
The discussion has been also revised and new evidence has been provided (check the text, from line 309 to 396).
8. Author Contributions. With Initials.
Performed.
Round 2
Reviewer 2 Report
Thank you to the authors for responding to all comments. The manuscript was improved. Some minor concerns remain.
- The reader may benefit from a bit more clarification about the statistical analyses. For instance, it says that they compared "one quantitative and one categorical variable" using Student t-test, but student t-tests are only used to compare continuous independent variables, and then there were multiple ways of controlling for multiple comparisons. It may help if they list the variables for each analysis. That is, for what analyses they used Bonferroni vs Games-Howell, etc.
- The authors now mention that 44 people were excluded because of "abnormal responses." Please clarify what that means.
- Please only leave the exact p value in the results section and remove "> p = .00416" as this is already mentioned in the methods section.
- Please consider using asterisks to denote statistical significance in the tables instead of highlighting .
Author Response
Thank you very much for all of your suggestions through the revision. We respond to the comments below:
1. The reader may benefit from a bit more clarification about the statistical analyses. For instance, it says that they compared "one quantitative and one categorical variable" using Student t-test, but student t-tests are only used to compare continuous independent variables, and then there were multiple ways of controlling for multiple comparisons. It may help if they list the variables for each analysis. That is, for what analyses they used Bonferroni vs Games-Howell, etc.
Response:
Thank you for your comment. We hope that now this point will be clearer.
We have changed the previous paragraph:
"We compared one quantitative and one categorical variable (with two groups) through Student’s t-tests. We used the robust Welch’s Test whether the assumption of homogeneity of variances (Levene’s Test) was violated. We used Chi-square test with the McNemar test correction for 2x2 tables and the Bonferroni correction for multiple testing."
For this one:
"Later, we compared men and women respect to different variables. Concretely, to analyze differences between groups respect to sociodemographic variables (measured as categorical) we used Chi-square test with the McNemar test correction for 2x2 tables and the Bonferroni correction for multiple testing. Later, we used Student’s t-tests for analyzing if measures of mental health (measured as quantitative) were significantly different in men and women. We adjusted for multiple comparison applying the Bonferroni correction and we used the robust Welch’s Test whether the assumption of homogeneity of variances (Levene’s Test) was violated."
Later, in the section of univariate general linear analyses, we have changed the paragraph too. We have changed the previous one:
"We analyzed the homogeneity variance assumption, and we considered the correction of Bonferroni or Games-Howell for multiple testing (depending on the homogeneity variance assumption was accepted or violated)."
For this one:
"We analyzed the homogeneity variance assumption, and we applied the correction of Bonferroni for multiple testing when the assumption of homogeneity variance was accepted or the correction of Games-Howell when the assumption was violated."
2. The authors now mention that 44 people were excluded because of "abnormal responses." Please clarify what that means.
Response:
We have included additional information to clarify this point:
"...and 44 abnormal responses (the more common was to provide the same response, for example 3, in all questions)."
3. Please only leave the exact p value in the results section and remove "> p = .00416" as this is already mentioned in the methods section.
Response:
Thank you. We have modified the p value accordingly with the text, considering that the p value of reference is corrected by Bonferroni (as we have explained in line 160). Concretely:
"Women reported higher scores than men, but non-significantly difference, regarding self-care patterns (t(1) = -2.14, p > .004, d = 0.08) and no positive activities subscale (t(1) = -2.19, p > .004, d = 0.08)."
The exact p values are in Table 2.
4. Please consider using asterisks to denote statistical significance in the tables instead of highlighting.
Response:
Thank you, we have changed it in Table 1 and 3.
Reviewer 4 Report
Thanks for adresse the changes that I suggested.
Please clarify Institutional Review Board Statement with the protocol Number if it aplica.
Author Response
Thank you very much for all of your suggestions through the revision. We respond to the comments below.
1. Please clarify Institutional Review Board Statement with the protocol Number if it aplica.
Response:
The ethical committee of Universidad Loyola Andalucía do not provide a protocol number (I attach it). Nevertheless, we have included in the manuscript the day in which we received the approval. Concretely:
"The study was conducted according to the guidelines of the Declaration of Helsinki, and approved by the Ethics Committee of Loyola Andalucía University (Spain) (25th March 2020 date of approval)."
